# ContraSim: Contrastive Similarity Space Learning for Financial Market Predictions

## Abstract

We introduce the Contrastive Similarity Space Embedding Algorithm (**ContraSim**), a novel framework for uncovering the global semantic relationships between daily financial headlines and market movements. ContraSim operates in two key stages: **(i)** Weighted Headline Augmentation, which generates augmented financial headlines along with a semantic fine-grained similarity score, and **(ii)** Weighted Self-Supervised Contrastive Learning (WSSCL), an extended version of classical self-supervised contrastive learning that uses the similarity metric to create a refined weighted embedding space. This embedding space clusters semantically similar headlines together, facilitating deeper market insights. Empirical results demonstrate that integrating ContraSim features into financial forecasting tasks improves classification accuracy from WSJ headlines by 7%. Moreover, leveraging an information density analysis, we find that the similarity spaces constructed by ContraSim intrinsically cluster days with homogeneous market movement directions, indicating that ContraSim captures market dynamics independent of ground truth labels. Additionally, ContraSim enables the identification of historical news days that closely resemble the headlines of the current day, providing analysts with actionable insights to predict market trends by referencing analogous past events.

## 1 Introduction

With the rapid advancements in the capabilities of Large Language Models (LLMs), researchers have significantly enhanced their ability to analyze and leverage the semantic richness of textual data for various downstream tasks. Mature fields such as Sentiment Analysis Devlin et al. [2019], Spam Detection Aggarwal et al. [2022], Machine Translation Vaswani et al. [2017], and many more Liu et al. [2019], Brown et al. [2020], Radford et al. [2019] have been completely revolutionized by the advent of deep LLMs. Likewise, because a key source of information in the domain of financial market movement prediction is encoded in textual representations (news, reports, social media, etc.), a predictable field of study has been how LLMs can be used to better predict market movement.

It is known that the direction of a stock's price is impacted by a plethora of temporally linked features, like overall market movement, industry trends and company-specific news. It has been a daunting task for researchers to build machine learning algorithms that are able to interpret the complex and noisy feature space of textual financial news, to repeatedly perform well in market movement prediction. Previous models created the majority of their predictive powers by solely looking at historic financial indicators Fischer & Krauss [2018], Sezer & Ozbayoglu [2018]. However, with LLM's ability to create dense feature representations from human text, composite models that utilize financial indicators in conjunction with news and social-media posts were able to improve predictive performance Saqur [2024], Liu et al. [2021]. Multiple projects have found success doing this by using a mixture of classical and deep learning approaches Ding et al. [2015],

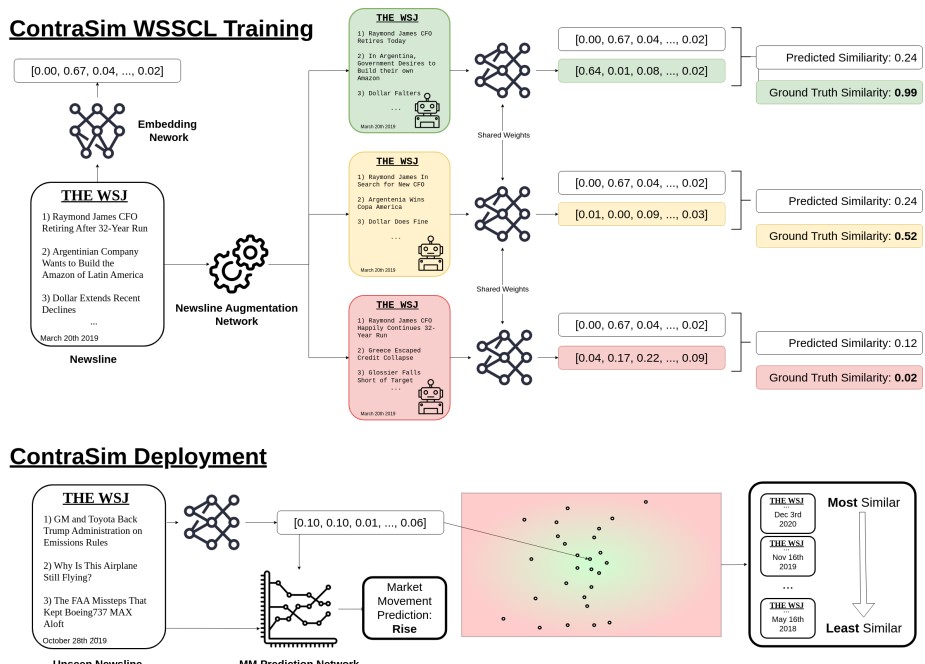

Figure 1: Overview of our proposed Contrastive Similarity (ContraSim) embedding approach. **In training**, we use a LLaMA chat model to generate augmented financial news headlines with varying degrees of semantic similarity to the original. We then use a Weighted Self-Supervised Contrastive Learning (WSSCL) approach to create an embedding space that clusters semantically similar prompts closer together. **In deployment**, the embeddings from the similarity space, can be used to i) Make better predictions on the direction of today's stock movement, ii) Find the most similar financial news to today's.

Fischer & Krauss [2018], Hu et al. [2018], Sezer & Ozbayoglu [2018], Xu et al. [2018], Liu et al. [2021]. State of the art approaches to stock market prediction is outlined in section 2.

While composite models that blend financial indicators with language features have improved market movement predictions, they often function as "black boxes." They predict market changes without offering any insight into why a particular prediction was made, making them less useful for financial analysts seeking interpretability. To address this, we propose a Contrastive Self-Supervised Learning approach that not only enhances market movement predictions using financial text data but also preserves interpretability. Our method aims to: a) predict the current day's market direction using Wall Street Journal (WSJ) headlines The Wall Street Journal [2024], and b) provide a ranked list of similar past financial news events.

The idea behind our approach is straightforward. We treat a day's news as a combined list of all WSJ (and other relevant, reputable sources) headlines for that day. For example, a headline like "*Canadian Crude Prices Hit by Keystone Pipeline Shutdown*" (2019-11-05) serves as input, much like other models. However, in addition to predicting market changes, our approach also identifies other days when similar events occurred. For instance, the most similar past headline might be "*Russian Pipeline Shutdown Shifts Balance in Oil Market*" (2019-05-22). This method offers a balance of interpretability and simplicity, allowing analysts to identify patterns in current news and historical contexts without relying on a complex "Explainable AI" (XAI) component.

We propose ContraSim, a method that leverages a novel textual augmentation algorithm powered by LLMs to generate diverse news headlines with varying degrees of semantic similarity to the original. Augmented headlines are assigned similarity scores ranging from 1.0 (high semantic alignment) to 0.0 (completely disjoint meaning). Using these augmented pairs, we introduce Weighted Self-Supervised Contrastive Learning (WSSCL) to build an embedding space where semantically similar headlines are naturally clustered. This embedding algorithm enables the calculation of similarity scores between any two real-world headlines based on their semantic proximity.

This approach is validated through two key findings: a) WSSCL inherently groups headlines associated with similar market directions closer in the embedding space. Even without explicit market movement labels, the model intuitively captures the relationship between headlines and market behavior using an information-gain framework, and b) a large language model (LLM) trained with WSSCL-enhanced embeddings outperforms an LLM relying solely on raw financial headlines for market movement prediction, demonstrating the added value of this semantic embedding strategy.

**Contributions** : We introduce the *Contrastive Similarity Space Embedding Algorithm* (ContraSim), a method that generates headline augmentations with meaningful and nuanced similarity coefficients. We demonstrate that:

a) ContraSim enables inter-day financial comparisons, allowing forecasters to identify historic market days similar to the current day.

b) ContraSim learns a mapping between news headlines and market direction in an unsupervised manner. This is evidenced by emergent structures in the embedding space that increase global insight into stock movement – i.e., by identifying similar prompts, we gain insight into why stocks move.

c) The similarity embedding spaces created by ContraSim enhance the performance of financial forecasting classification algorithms when used together.

**Organization:** Section §2 reviews the foundational concepts and situates our work within the existing literature. Section §3 describes the proposed methodologies in detail. Section §4 presents our experimental setup, empirical findings, and a discussion of training details, along with directions for future research. Additional details, including a comprehensive explanation of headline transformations (§A), examples of similar news headlines (§B), an analysis of how augmentation actions influence similarity (§C), and dataset descriptions (§D), are provided in the appendix.

## 2 RELATED WORKS

**Machine Learning in Financial Forecasting** Early approaches to predicting stock market movements relied heavily on classical statistical models. One foundational method, the Autoregressive Integrated Moving Average (ARIMA) Box & Jenkins [1970], utilized time series data to forecast trends. Subsequent models, such as *Generalized Autoregressive Conditional Heteroskedasticity* (GARCH) Bollerslev [1986], *Vector Autoregression* (VAR) Sims [1980], and *Holt-Winters exponential smoothing* Holt [1957], extended these capabilities by capturing more intricate patterns in financial time series. Other notable contributions include techniques for cointegration analysis Engle & Granger [1987], Kalman filtering Kalman [1960], and Hamilton's regime-switching models Hamilton [1989].

While effective, these classical models were primarily limited to tabular datasets and struggled with nonlinear relationships and multimodal inputs. The rise of Large Language Models (LLMs) transformed financial forecasting by enabling the incorporation of richer, more complex data sources. For example, integrating financial news articles Yang et al. [2020], sentiment analysis Yang et al. [2020], social media activity Bollen

et al. [2011], and earnings call transcripts Tsai & Wang [2016] significantly enhanced market movement predictions, demonstrating the versatility and power of LLMs in handling diverse financial modalities.

**Contrastive Learning**   Contrastive learning has emerged as a powerful paradigm in unsupervised and self-supervised learning, focusing on representation learning through comparisons. The core idea is to bring similar data points closer in the representation space while distancing dissimilar ones. A key milestone in this field was SimCLR Chen et al. [2020], which used data augmentations and contrastive loss to learn high-quality representations without requiring labels. MoCo He et al. [2020] further advanced this approach by introducing a memory bank to efficiently manage negative examples, making it more scalable for larger datasets. Recent innovations like SimSiam Chen & He [2021] have shown that competitive representations can be learned without relying on negative pairs, streamlining computation and improving accessibility. These advancements are particularly relevant for financial applications, where large-scale and heterogeneous datasets are common, enabling contrastive learning to uncover nuanced relationships in financial data.

## 3   METHODS

In this section, we introduce ContraSim, a self-supervised contrastive learning algorithm that creates augmented news headlines with fine-grained degrees of similarity to the base. Then using a weighted self-supervised learning paradigm, we create an embedding space, where semantically similar news headlines are clustered together. Additionally, we outline how we can measure the efficacy of ContraSim by using an information density approach in our similarity space to see if there is inherent market-movement knowledge being learned by optimizing for news headline similarity.

### 3.1   CONTRASIM: CONTRASTIVE SIMILARITY SPACE EMBEDDING ALGORITHM

Here, we formulate the news headline augmentation pipeline and the Weighted Self-Supervised Contrastive Learning (WSSCL) approach that in tandem generate the ContraSim. The contrastive similarity space, generated from ContraSim, is optimized to put the headlines with semantically similar news into local proximity.

We define the Daily-News Set (DNS) dataset as:

$$\mathcal{D}_{\text{DNS}} = \{(d_i, \mathcal{N}_i) \mid i = 1, 2, \ldots, n\}, \quad \text{where } \mathcal{N}_i = (h_{i1}, h_{i2}, \ldots, h_{im}) \tag{1}$$

Where, $n$ is the total number of news headlines within the news headline dataset, $\mathcal{N}_i$ is news headline object containing a tuple of headlines strings $h$, and $d_i$ is the corresponding date identifier string for a day $i$.

In this context, a **Daily News-Set (DNS)** is a collection of WSJ The Wall Street Journal [2024] headlines from a specific day. Later in this paper we explore using Social Media posts to form a DNS, and also explore how well ContraSim performs on other non-financial textual domains (e.g. list of movie reviews).

**1. Defining the Augmentation Objective**   Below, we propose a stochastic transformation $T : \mathcal{N} \to (s, \hat{\mathcal{N}})$, where $\mathcal{N}$ is an input DNS, $\hat{\mathcal{N}}$ is the augmented DNS, and $s \in [0.0, 1.0]$ represents the similarity score between $\mathcal{N}$ and $\hat{\mathcal{N}}$. In subsection 3. we further discuss our implementation details and our process of measuring inter-DNS semantic similarity.

The dominant strategy for creating contrastive embedding spaces defines inter-object relationships in binary terms: two objects are either within the same class or outside the same class. However, for this objective, we do not have access to binary class labels between daily news sets, as the similarity between daily news sets is inherently continuous and varies along a continuous spectrum. Weighted contrastive approaches, such as Xi

et al. [2022], better align with this setting by leveraging nuanced similarity scores to guide the embedding space construction, enabling more accurate representation of the semantic relationships between augmented DNS.

**2. Generating Augmented Daily News Sets**    Augmented DNS are generated through the following discrete actions: i) Rewording an original headline (**Re**), ii) Generating a semantically shifted version (**S**), iii) Negating an original headline (**N**), and iv) Selecting a random headline from a different day (**Ra**).

To achieve these transformations, we leveraged the LLaMA-3-7b-chat model Touvron et al. [2023], prompting it with carefully crafted instructions tailored to each specific action. For rewording (**Re**), the model was prompted to retain the original meaning of the headline while rephrasing it with alternative wording and sentence structure. For semantic-shifting (**S**), the prompt instructed the model to subtly alter the meaning of the headline, introducing slight semantic deviations while maintaining topical relevance. For negation (**N**), the model was guided to generate a headline that conveyed the direct opposite meaning of the original. By using these tailored prompts, the LLaMA model provided high-quality augmented news headlines that covered a broad spectrum of semantic variations.

To ensure the quality of LLM-generated headline augmentations, we employ an off-the-shelf fine-tuned BERT model as a discriminator to verify semantic consistency. This model takes the base and augmented headlines as inputs and outputs a semantic similarity score, bounded between 0 and 1. The score thresholds define stricter guidelines for each augmentation action, where negated, semantically-shifted, and reworded headlines must fall in ranges (0, 0.33), (0.33, 0.66), and (0.66, 1.00) respectively. This approach provides a well-defined, quantitative framework for categorizing augmentations. It not only enforces consistency in semantic relationships but also ensures that the augmented headlines are reliable and aligned with the intended transformations.

A further exploration on the specifics of the three (steps (i)-(iii)) headline transformations are expanded upon in appendix section A. Table 1 depict a pedagogical example illustrating these transformations:

| Transformation Action | Example Headline |
|---|---|
| **Original** | *Johnson & Johnson to Buy Surgical Robotics Maker Auris* |
| **Reworded (Re)** | *Auris Acquired by Pharmaceutical Giant Johnson & Johnson* |
| **Semantically-Shifted (S)** | *Abbott Laboratories Acquires Medical Imaging Specialist Siemens Healthineers* |
| **Negated (N)** | *Auris to Sell Off Stake in Surgical Robotics Business to Johnson & Johnson* |

Table 1: Example transformations of a news headline using the LLaMA-3-7b-chat model.

The final augmentation action **Ra**, is a function that randomly selects a headline from the training split (ignoring headlines within the base DNS $\mathcal{N}$). This acts similarly to randomly sampling negatives in a traditional contrastive learning mechanism.

Our augmentation stochastic transformation $T : \mathcal{N} \to (s, \hat{\mathcal{N}})$, generates augmented DNS defined fully in Appendix A. However, the intuition is quite straightforward. For each DNS, we can generate an augmentation by: 1) Determine the number of headlines in the augmented DNS ($\hat{\mathcal{N}}$) by sampling from the global distribution 2) For each augmented headline $\in \hat{\mathcal{N}}$, randomly sample an augmentation action from $P_{actions}$ 3) Perform the sampled augmentation action. Note that the actions **Re**, **S**, and **N** each randomly sample a headline from the base DNS ($\mathcal{N}$), and use that to create an augmented headline. 4) Randomly shuffle the order of the augmented headlines in $\hat{\mathcal{N}}$.

In our experiments we set $P_{actions}$ such that: $P(\mathbf{Re}) = 0.05$, $P(\mathbf{S}) = 0.025$, $P(\mathbf{N}) = 0.05$, and $P(\mathbf{Ra}) = 0.775$. These values were used because augmented DNS produced a similarity score distribution with a high skew to negative scores (as common in many contrastive learning frameworks), while also not overly depending on negative action augmentations. We leave finetuning this probability distribution as a task for future work.

**3. Generating Similarity Scores**  For each augmented news headline $\hat{\mathcal{N}}$, we calculate the similarity score $S(\hat{\mathcal{N}})$ using a logarithmic weighting function:

$$S(\hat{\mathcal{N}}) = \ln\left(1 + \frac{\sum_{a \in \hat{\mathcal{N}}_A} Sim(a)}{S_{\max}} \cdot (e - 1)\right) \tag{2}$$

where $a$ is an augmentation action within the list of augmentation action tuple $\hat{\mathcal{N}}_A$, $S_{\max}$ is the maximum possible total score to normalize the sum to the range $[0, 1]$, and $Sim(.)$ is the function mapping each augmentation action to its corresponding similarity score, such that:

$$Sim(\mathbf{Re}) = 1.0, \quad Sim(\mathbf{S}) = 0.5, \quad Sim(\mathbf{N}) = 0.0, \quad Sim(\mathbf{Ra}) = 0.0$$

**Intuition**: The goal of generating a similarity score is to create a metric between 0.0 and 1.0 that measures how similar a DNS is semantically to its augmentation. When comparing two headlines, we assign high similarity if they are rephrased but semantically identical to each other (**Re**), medium similarity if they are semantically-shifted (**S**), and low similarity if they are semantic opposites (**N**) or completely different (**Ra**).

A simple approach to generating a similarity scores between a DNS and its augmentation could be to take the simple mean of all of the augmentation action scores. However, if we observe that two DNS each have a headline that is semantically identical but just reworded, then we want to take note that those DNS are so similar. Equation 2, skews the similarity scores such that

| | **Re** | **S** | **N** | **Ra** | $S(\hat{\mathcal{N}})$ |
|---|---|---|---|---|---|
| $\hat{\mathcal{N}}_1$ | 15 | 1 | 0 | 15 | 1.00 |
| $\hat{\mathcal{N}}_2$ | 5 | 3 | 1 | 21 | 0.53 |
| $\hat{\mathcal{N}}_3$ | 1 | 4 | 4 | 17 | 0.29 |
| $\hat{\mathcal{N}}_4$ | 0 | 0 | 1 | 26 | 0.00 |

Table 2: List of augmentation actions from a base daily news set, and their accompanying similarity score.

actions with higher similarity scores have an exponentially larger effect in DNS similarity, than semantically different actions. An example of similarity scores is outlined in table 2. There, we see that if we have an augmented DNS, $\hat{\mathcal{N}}_\infty$, that has 15 semantically identical headlines to the base DNS, then the similarity score should be very high. Furthermore, $\hat{\mathcal{N}}_\triangle$ is a headline with one semantically negated headline from the original, and the rest are completely disjoint headlines, and so it has a very low semantic similarity.

## 3.2 WEIGHTED SELF-SUPERVISED CONTRASTIVE LEARNING (WSSCL)

With augmented daily news sets (DNS) generated from the training set, and similarity scores assigned to each anchor-augmentation DNS pair, we can now proceed to construct the similarity embedding space using a weighted self-supervised contrastive learning approach.

Our embedding space optimization task is inspired by Supervised Contrastive Learning Khosla et al. [2021], but is augmented to allow for regressive similarity measurements between anchor and augmented projections instead of binary positive / negative labels. Our representation learning framework consists of 3 components, the **Encoder Network**, the **Projection Network**, and the **Classification Networks**:

**Encoder Network**: $e = Enc(x)$ is a LLaMA-3 AI [2024] 7 billion parameter chat model. It was fine-tuned to predict market movement direction (*Fall*, *Neutral*, or *Rise*) from the NIFTY-SFT dataset Raeid et al. [2024].

Additional details of SFT implementation are available from Saqur [2024]. Concatenated daily news sets are tokenized and propagated through the encoder network, and the mean values from the last hidden layer are returned, such that $e = \text{Enc}(x) \in \mathbb{R}^{D_E}$. $e$ is then normalized to a hypersphere, which in our implementation had dimensions of 4096.

**Projection Network**: $p = Proj(e)$ is a feedforward neural network with a single hidden layer, and a shape of (4096, 256, 128), and a single ReLU nonlinearlity unit. The role of this network is to project embeddings $e$ into our embedding space. After projection, the output values are again normalized. We found negligible effects on the quality of the embedding space by increasing the complexity of the projection network.

**Classification Networks**: $Class_{Proj}(p)$, $Class_{LLM}(x)$ and $Class_{Both}(p, x)$, are tasked with classifying the market movement as rising, falling or neutral. $Class_{Proj}$ takes the projections from the embedding space as an input and $Class_{SFT}$ takes the final hidden states from a separate LLM finetuned for market prediction. $Class_{Both}(p, x)$ takes both projection and SFT embeddings as inputs. Training of the classification networks is done after the projection network is optimized. Note that for training of the classification networks all augmentations are discarded, and our classifiers are optimized on real news headlines only.

The optimization task we define for our projection network are defined the Weighted Similarity Contrastive Loss (Equation 3).

$$\mathcal{L}_{\text{WSCL}} = \frac{1}{|\mathcal{D}_{newsheadlines}|} \sum_{i=1}^{N} \sum_{j=1}^{M_i} \left[ s_{ij} \cdot d_{ij}^2 + (1 - s_{ij}) \cdot \max(0, \delta - d_{ij})^2 \right], \tag{3}$$

Where, $N$: Total number of anchor news headlines in a batch, $M_i$: Number of augmented samples for anchor $i$, $d_{ij} = ||\mathbf{p}_i - \mathbf{q}_{ij}||_2$, $s_{ij} \in [0, 1]$: Similarity score between the anchor and augmented embeddings, and $\delta$ is the hyperparameter defining the contrastive margin.

The proposed loss ($\mathcal{L}_{\text{WSCL}}$) extends the classical triplet loss by incorporating a fuzzy similarity score $s_{ij} \in [0, 1]$, enabling a more nuanced handling of relationships between anchor and augmented samples. This formulation draws inspiration from the traditional triplet loss introduced by Schroff et al. [2015]. in FaceNet, which minimizes the distance between anchor-positive pairs while maximizing the distance between anchor-negative pairs using a fixed margin. By replacing binary labels with continuous similarity values, $\mathcal{L}_{\text{WSCL}}$ facilitates a finer gradient flow and captures graded relationships, making it particularly suitable for tasks involving regressive or weighted similarity measures.

The **pull loss** term, $s_{ij} \cdot d_{ij}^2$, minimizes the distance between anchor and augmented embeddings when $s_{ij}$ is high (e.g., $s_{ij} \approx 1.0$). Conversely, the **push loss** term, $(1 - s_{ij}) \cdot \max(0, \delta - d_{ij})^2$, increases the distance between embeddings when $s_{ij}$ is low (e.g., $s_{ij} \approx 0.0$), ensuring proper separation within the embedding space.

In addition to $\mathcal{L}_{\text{WSCL}}$, the Continuously Weighted Contrastive Loss (CWCL) proposed by Srinivasa et al. [2023] is another approach for weighted similarity learning. Unlike $\mathcal{L}_{\text{WSCL}}$, CWCL uses cosine similarity instead of Euclidean distance and incorporates a softmax normalization across all pairs in the batch to enforce global consistency. The CWCL loss is defined as:

$$\mathcal{L}_{\text{CWCL}} = -\frac{1}{|\mathcal{D}_{newsheadlines}|} \sum_{i=1}^{N} \sum_{j=1}^{M_i} s_{ij} \cdot \log \frac{\exp(-d_{ij}/\tau)}{\sum_{k=1}^{M_i} \exp(-d_{ik}/\tau)}, \tag{4}$$

Where $\tau$ is the temperature scaling parameter that controls the sharpness of the distribution. CWCL allows for fine-grained alignment of embeddings by normalizing similarity scores within the batch, providing a complementary perspective to the pull-push mechanics of $\mathcal{L}_{\text{WSCL}}$.

Both approaches aim to improve the representation of graded relationships in embedding spaces but differ in their distance metrics and weighting strategies. In Section 4, we explore each loss function and measure which one performs better on our evaluation tasks.

It is notable that for the WSSCL task, the ground truth market direction corresponding to the DNS's day is not used at all in clustering. The ground truth market direction is saved only for our evaluation tasks (see subsection 3.3). This is so we can measure if the self-supervised task, optimized only for similarity inherently encodes market direction features, without giving them specifically. This lends credence to the idea that through WSSCL information on markets is created.

### 3.3 EVALUATING SIMILARITY SPACE INFORMATION RICHNESS

To measure the efficacy of ContraSim, we employ two approaches. The first is the most straightforward: we train a market movement prediction algorithm using both ContraSim embeddings and a baseline without ContraSim embeddings, and evaluate its downstream classification performance.

The second approach involves analyzing how our projection network ($Proj(e)$) embeds real-world daily news sets (DNS). Using information-dense metrics, we evaluate whether the model inherently clusters DNS associated with the same market direction closer together. For instance, if the similarity space places headlines corresponding to rising markets near one another, it suggests that meaningful information is being captured. This clustering behavior is quantified using four information-dense metrics:

**1) Geometric K-Nearest Neighbors (g-KNN):** This metric evaluates the quality of local label distributions by measuring the entropy of the labels among the $k$-nearest neighbors of each data point, averaged over the dataset. It provides insights into the local clustering structure of the embedding space Lord et al. [2018]. **2) Nearest Neighbor Accuracy:** This metric assesses the proportion of data points whose closest neighbor shares the same category label, offering a direct measure of clustering performance. **3) Kullback-Leibler (KL) Divergence:** This metric quantifies the difference between the local label distribution among the $k$-nearest neighbors and the global label distribution, highlighting the extent to which local clusters deviate from random chance Shlens [2014]. **4) Jensen-Shannon Divergence (JSD):** This symmetric and bounded metric evaluates the similarity between local and global label distributions, enhancing interpretability. It is widely recognized for its effectiveness in quantifying clustering quality and information richness in embedding spaces Lin [1991].

## 4 EXPERIMENTAL RESULTS AND INTERPRETATIONS

In this section, we evaluate the performance and interpretability of the ContraSim framework across multiple datasets and tasks. The experiments are designed to assess both the downstream classification capabilities of ContraSim embeddings and the inherent clustering properties of the generated similarity space. By testing on datasets from diverse domains—financial news (NIFTY-SFT, BigData22) and movie reviews (IMDB)—we aim to demonstrate the generalizability of ContraSim beyond financial prediction tasks. Additionally, we utilize a range of quantitative metrics, including accuracy, F1 score, and embedding space density metrics, to measure the quality and effectiveness of the embeddings. These evaluations provide insights into the practical utility of ContraSim for supervised learning and its ability to create meaningful representations that capture domain-specific nuances.

### 4.1 DATASETS

For each of these experiments, we compare results on 3 datasets: NIFTY-SFT Saqur et al. [2024], BigData22 Soun et al. [2022], and the IMDB review dataset Maas et al. [2011]. A full analysis of this is outlined in Table 3. **NIFTY-SFT** Saqur et al. [2024] is the collection of WSJ headlines The Wall Street Journal [2024]

collected and concatenated together alongside the movement of the US equities market (ticker: $SPY) for the corresponding day. **BigData22** Soun et al. [2022] likewise is a financial news headline dataset, but news headlines are composed of tweets as apposed to WSJ headlines. Finally, we evaluate with the **IMDB review** dataset, which is a collection of human-written reviews for a list of movies alongside the movie's overall review score. An extended analysis of the datasets used is available in Appendix D.

For the IMDB review example, we define a news headline as the concatenated movie reviews, and the prediction task into *Low* (0.0 - 5.5 stars), *Medium* (5.6 - 7.5 stars) and *High* (7.6 - 10.0 stars). We evaluate ContraSim on this dataset to assess its generalizability to orthogonal tasks beyond financial domain prediction.

| Dataset | Problem Domain | Headlines/Reviews | Days/Movies | Date Range |
|---|---|---|---|---|
| NIFTY-SFT | Financial Headlines | 18,746 | 2,111 | 2010/01/06-2017/06/27 |
| BigData22 | Financial Tweets | 272,762 | 7,164 | 2019/07/05, 2020/06/30 |
| IMDB Review | Movie Reviews | 50,000 | 1,000 | 1874, 2020 |

Table 3: Summary of the datasets used in the experiments, including their problem domain, the number of headlines, the number of days, and the date range.

## 4.2 RESULTS

Table 4 demonstrates that combining similarity space projections with LLM embeddings improves the classification of news headlines into rising, neutral, or falling categories. Specifically, applying this conjunctive approach to the NIFTY-SFT dataset results in a balanced accuracy of 37.79%, reflecting a 13% increase over the baseline and a 7% improvement compared to using only LLM embeddings. In contrast, the model trained exclusively on projections performed slightly better than the baseline. Similarly, on the IMDB dataset, the composite model outperformed the baseline LLM, achieving a 6.8% increase in

| Model | NIFTY-SFT | BigData22 | IMDB |
|---|---|---|---|
| Baseline | .3333 / .3333 | .5000 / .5000 | .3333 / .3333 |
| $Class_{CWCL}$ | .3512 / .3433 | .5005 / .5016 | .3900 / .3897 |
| $Class_{WSCL}$ | .3505 / .3336 | .5014 / .5019 | .4044 / .3992 |
| $Class_{LLM}$ | .3522 / .3833 | .5150 / **.5094** | .4518 / .4124 |
| $Class_{LLM+CWCL}$ | **.3779** / **.3712** | .5156 / .5089 | **.5198** / **.4620** |
| $Class_{LLM+WSCL}$ | .3678 / .3680 | **.5167** / .5090 | .5103 / .4498 |

Table 4: Accuracies and F1 scores (**Accuracy / F1 Score**) for classification models across the three datasets. The NIFTY-SFT and IMDB datasets were subsetted to achieve a (33%, 33%, 33%) split. The BigData22 dataset with only *Fall* and *Rise* labels was subsetted to (50%, 50%). Best results and approximately equal to best are in **bold** and underline respectively.

accuracy and a 0.0496 improvement in F1 score. However, for the Bigdata22 dataset, no significant differences in accuracy or F1 score were observed between the models. Similar performance was found across each of the losses.

Table 5 presents embedding space density metrics for the baseline model and our similarity space projections, evaluated across three datasets. The results demonstrate that the ContraSim embedding space, optimized through WSSCL and CWCL losses, consistently outperforms the baseline in g-KNN, KNN, KL-Divergence, and JSD metrics. Notably, the $\mathcal{L}_{WSCL}$ projection achieves the highest g-KNN and KNN scores on the NIFTY-SFT dataset, indicating better local neighborhood density and improved separability in the embedding space. Similarly, $\mathcal{L}_{WSCL}$ and $\mathcal{L}_{CWCL}$ models are competitive, with $\mathcal{L}_{WSCL}$ excelling in KL-Divergence and JSD scores on BigData22, suggesting enhanced distributional alignment.

These results provide strong evidence that the WSSCL process inherently generates informative market representations without requiring ground truth labels. Moreover, the competitive performance of the $\mathcal{L}_{\text{WSCL}}$ models across datasets underscores their ability to generalize across diverse textual domains, reinforcing the utility of similarity space projections for various tasks.

| Dataset | Model | g-KNN (k=5) ($\uparrow$) | KNN (k=5) ($\uparrow$) | KL-Divergence ($\uparrow$) | JSD ($\uparrow$) |
|---|---|---|---|---|---|
| **NIFTY-SFT** | Baseline | .5916 | .4668 | .3539 | .1054 |
| | $\mathcal{L}_{\text{CWCL}}$ | .7647 | .4732 | **.3821** | **.1164** |
| | $\mathcal{L}_{\text{WSCL}}$ | **.7219** | **.5205** | .3740 | .1144 |
| **BigData22** | Baseline | .7951 | .5506 | .1499 | .0452 |
| | $\mathcal{L}_{\text{CWCL}}$ | **.9084** | **.7101** | .2030 | .0607 |
| | $\mathcal{L}_{\text{WSCL}}$ | .8590 | .5507 | **.2246** | **.0640** |
| **IMDB** | Baseline | .7456 | .5781 | .2919 | .0818 |
| | $\mathcal{L}_{\text{CWCL}}$ | .7626 | **.7500** | **.3957** | **.1120** |
| | $\mathcal{L}_{\text{WSCL}}$ | **.8252** | .6875 | .3024 | .0908 |

Table 5: Comparison of Baseline and Projection models across datasets and evaluation metrics. Note that finding true baseline values for these metrics on unbalanced sets of labels is nontrivial and out of scope for this paper. As a result, estimated baseline values are the mean of 1000 cases of randomly distributed points following the respective label splits for each dataset. The best results are in **bold**.

### 4.3 TRAINING DETAILS

The projection network was trained for 50 epochs using $\mathcal{L}_{\text{CWCL}}$ and $\mathcal{L}_{\text{WSCL}}$ losses, with a learning rate of 0.001 and a batch size of 2. We optimized using the Adam optimizer ($\beta_1 = 0.9, \beta_2 = 0.999$). A cosine annealing schedule was applied to adjust the learning rate, and gradient clipping with a norm of 1.0 ensured training stability. The datasets were split into 80% training, 10% validation, and 10% test sets, and augmentation probabilities were tuned to maximize similarity learning.

## 5 FUTURE WORK

Future research should explore applying ContraSim to diverse domains such as healthcare, legal, and social media datasets to evaluate its generalizability across varying text types and contexts. Additionally, incorporating advanced LLMs like GPT-4 may enhance embedding quality and clustering performance. Investigating the integration of techniques such as hard negative mining, dynamic temperature scaling, and multi-task learning with ContraSim could further refine its representation capabilities. Lastly, extending ContraSim to real-time financial forecasting applications and unsupervised learning scenarios may yield insights into dynamic market behavior.

**Reproducibility Statement** The authors of this paper ensure reproducibility through 1) The accurate and clear descriptions of methods used, specifically in the training details and methods sections of the text, 2) The Use of only public models and datasets (NIFTY), and 3) Providing source code in the supplemental materials (see attached).

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

# Appendices

## Appendix Contents

# A HEADLINE TRANSFORMATIONS

## A.1 OUTLINING THE HEADLINE TRANSFORMATION ALGORITHM

In Algorithm 1, we outline the steps for generating augmented DNS, from the corpus of news headlines. Note that the details of the augmentation actions are shown in subsection A.2.

---

**Algorithm 1:** Stochastic Daily News Set Augmentation Transformation $T$

---

**Input:** Original news headline $\mathcal{N} = (h_1, h_2, \ldots, h_m)$
**Input:** Action distribution $P_{\text{actions}}$ over actions $\{\mathbf{Re}, \mathbf{S}, \mathbf{N}, \mathbf{Ra}\}$
**Output:** Augmented news headline $(\hat{\mathcal{N}}, s)$ with similarity score $s$

1   Sample $n \sim$ Distribution of news headline lengths in corpus
2   Initialize $\hat{\mathcal{N}} \leftarrow \emptyset$, $S \leftarrow 0$
3   **for** $i \leftarrow 1$ **to** $n$ **do**
4      Sample $a_i \sim P_{\text{actions}}$
5      **if** $a_i \in \{\boldsymbol{Re}, \boldsymbol{S}, \boldsymbol{N}\}$ **then**
6         Sample headline $h \sim \mathcal{N}$
7      **else if** $a_i = \boldsymbol{Ra}$ **then**
8         Sample random headline $h \sim$ corpus
9      **end if**
10      **if** $a_i = \boldsymbol{Re}$ **then**
11         $h' \leftarrow \text{Reword}(h)$
12         $S \leftarrow S + 1.0$
13      **else if** $a_i = \boldsymbol{S}$ **then**
14         $h' \leftarrow \text{SemanticShift}(h)$
15         $S \leftarrow S + 0.5$
16      **end if**
17      **else if** $a_i = \boldsymbol{N}$ **then**
18         $h' \leftarrow \text{Negate}(h)$
19         $S \leftarrow S + 0.0$
20      **end if**
21      **else if** $a_i = \boldsymbol{Ra}$ **then**
22         $h' \leftarrow h$
23         $S \leftarrow S + 0.0$
24      **end if**
25      Append $h'$ to $\hat{\mathcal{N}}$
26   **end for**
27   Shuffle $\hat{\mathcal{N}}$
28   Compute similarity score $s \leftarrow S(N)$
29   **return** $(\hat{\mathcal{N}}, s)$

---

## A.2 HEADLINE AUGMENTATION

In this section, we outline the steps of generating the headline augmentations. In Tables 6, 7, and 8, we outline the prompts used to generate the reworded, semantically-shifted, and negated augmentations. Then, we show the quality control mechanism used to ensure semantic alignment between desired semantic similarity and actual semantic similarity of augmented headlines.

| | |
|---|---|
| | **Original Headline** |
| | "Johnson & Johnson to Buy Surgical Robotics Maker Auris" |
| | **Reworded Augmentation** |
| **System** | *Please reword this headline for me, preserving the exact semantic meaning perfectly. Your returned headline should contain the exact information with no meaning added or subtracted, but just rephrased. Please generate the headline, and return only that with no other text. Thanks.* |
| **User** | "Johnson & Johnson to Buy Surgical Robotics Maker Auris" |
| **Responses** | 1. Auris Acquired by Pharmaceutical Giant Johnson & Johnson |
| | 2. Auris Acquired by Pharmaceutical Giant Johnson & Johnson in Deal Involving Surgical Robot Technology |
| | 3. Auris Acquired by Pharmaceutical Giant Johnson & Johnson in Deal Involving Surgical Robot Manufacturer |
| | 4. Auris Manufacturer of Surgical Robots Acquired by Johnson & Johnson |
| | 5. Auris Medical Acquired by Pharmaceutical Giant Johnson & Johnson in Deal Involving Surgical Robotics Manufacturer |
| | **Semantic-Shift Augmentation** |
| **System** | *Please modify this headline slightly, so it is about something related but different. If the headline is good news, ensure it remains good news, and if it is bad news, ensure it remains bad news. Please generate the headline, and return only that with no other text. Thanks.* |
| **User** | "Johnson & Johnson to Buy Surgical Robotics Maker Auris" |
| **Responses** | 1. Abbott Laboratories Acquires Medical Imaging Specialist Siemens Healthineers |
| | 2. J&J Acquires Medtech Firm Verb for Improved Cancer Treatment Options |
| | 3. J&J Acquires Orthopedic Implant Company Zimmer Biomet in $14 Billion Deal |
| | 4. J&J Acquires Orthopedic Implant Company DePuy Synthes for $21 Billion |
| | 5. J&J Acquires Medtech Company Verb Surgical in $3 Billion Deal |
| | **Negative Augmentation** |
| **System** | *Please reword this headline for me such that the information is the same except that it now is about the opposite meaning. Please generate the headline, and return only that with no other text. Thanks.* |
| **User** | "Johnson & Johnson to Buy Surgical Robotics Maker Auris" |
| **Responses** | 1. Auris to Sell Off Stake in Surgical Robotics Business to Johnson & Johnson |
| | 2. Auris Abandons Plans to Acquire Surgical Robot Business from Johnson & Johnson |
| | 3. Auris to Sell Majority Stake to Rival of Johnson & Johnson's Surgical Robot Division |
| | 4. Auris Acquires Surgical Robotics Leader Johnson & Johnson |
| | 5. Auris Abandons Plans to Acquire Surgical Robotics Giant Johnson & Johnson |

Table 6: Rephrasing, slight ablation, and negative modification of the headline "Johnson & Johnson to Buy Surgical Robotics Maker Auris." Each augmentation displays the system prompt, user-provided headline, and model-generated responses listed with numbers.

|  | **Original Headline**
"Adviser Profile: Easing Client Worries in Uncertain Times" |
|---|---|
| **System** | **Reworded Augmentation**
*Please reword this headline for me, preserving the exact semantic meaning perfectly. Your returned headline should contain the exact information with no meaning added or subtracted, but just rephrased. Please generate the headline, and return only that with no other text. Thanks.* |
| **User**
**Responses** | "Adviser Profile: Easing Client Worries in Uncertain Times"
1. Finding Calm Amidst Turmoil: A Trustworthy Adviser's Approach to Alleviating Clients' Concerns
2. Managing Fears and Concerns of Clients Amidst Unpredictable Circumstances
3. Managing Client Anxiety Amidst Unpredictable Circumstances
4. Navigating Client Concerns Amid Unpredictable Circumstances
5. Managing Clients' Concerns Amidst Unpredictable Circumstances |
| **System** | **Semantic-Shift Augmentation**
*Please modify this headline slightly, so it is about something related but different. If the headline is good news, ensure it remains good news, and if it is bad news, ensure it remains bad news. Please generate the headline, and return only that with no other text. Thanks.* |
| **User**
**Responses** | "Adviser Profile: Easing Client Worries in Uncertain Times"
1. Financial Planner Profile: Boosting Clients' Confidence Amid Market Volatility
2. Financial Planner Spotlight: Calming Investor Fears Amid Market Volatility
3. Financial Planner Spotlight: Boosting Investor Confidence Amid Market Volatility
4. Financial Planner Spotlight: Offering Calming Expertise Amid Market Volatility
5. Financial Planner Focuses on Building Trust Amid Market Volatility |
| **System** | **Negative Augmentation**
*Please reword this headline for me such that the information is the same except that it now is about the opposite meaning. Please generate the headline, and return only that with no other text. Thanks.* |
| **User**
**Responses** | "Adviser Profile: Easing Client Worries in Uncertain Times"
1. Adviser Profile: Exacerbating Client Fears in Turbulent Eras
2. Adviser Profile: Exacerbating Client Fears in Chaotic Situations
3. Adviser Profile: Exacerbating Client Fears in Unpredictable Circumstances
4. Adviser Profile: Exacerbating Client Fears in Certain Circumstances
5. Adviser Profile: Exacerbating Client Fears in Turbulent Eras |

Table 7: Rephrasing, slight ablation, and negative modification of the headline "Adviser Profile: Easing Client Worries in Uncertain Times." Each augmentation displays the system prompt, user-provided headline, and model-generated responses listed with numbers.

| | **Original Headline** |
| --- | --- |
| | "U.S. Oil Producers Find a Surprise New Market: China" |

| | **Reworded Augmentation** |
| --- | --- |
| **System** | *Please reword this headline for me, preserving the exact semantic meaning perfectly. Your returned headline should contain the exact information with no meaning added or subtracted, but just rephrased. Please generate the headline, and return only that with no other text. Thanks.* |
| **User Responses** | "U.S. Oil Producers Find a Surprise New Market: China" |
| | 1. China Becomes an Unexpected Customer Base for U.S. Oil Producers |
| | 2. China Becomes an Unexpected Buyer of U.S. Crude Oil Supplies |
| | 3. China Emerges as an Unexpected Buyer of U.S.-Produced Crude Oil |
| | 4. China Becomes an Unexpected Buyer of U.S. Crude Oil Supplies |
| | 5. China Becomes an Unexpected Customer for U.S. Oil Producers |

| | **Semantic-Shift Augmentation** |
| --- | --- |
| **System** | *Please modify this headline slightly, so it is about something related but different. If the headline is good news, ensure it remains good news, and if it is bad news, ensure it remains bad news. Please generate the headline, and return only that with no other text. Thanks.* |
| **User Responses** | "U.S. Oil Producers Find a Surprise New Market: China" |
| | 1. U.S. Farmers Discover Surprising Boost in Exports to Asia |
| | 2. U.S. Farmers Reap Unexpected Benefits from Growing Trade Ties with China |
| | 3. China's Energy Crisis Eases as U.S. Oil Suppliers Step Up Deliveries |
| | 4. U.S. Farmers Discover Unexpected Export Boom in Asia |
| | 5. U.S. Natural Gas Exporters Strike Gold in Growing Asian Demand |

| | **Negative Augmentation** |
| --- | --- |
| **System** | *Please reword this headline for me such that the information is the same except that it now is about the opposite meaning. Please generate the headline, and return only that with no other text. Thanks.* |
| **User Responses** | "U.S. Oil Producers Find a Surprise New Market: China" |
| | 1. China's Oil Consumers Struggle to Fill Gap as U.S. Supplies Dry Up |
| | 2. China's Oil Demand Plummets, U.S. Producers Struggle to Fill Surplus Capacity |
| | 3. China's Oil Consumption Plummets, Leaving U.S. Producers Scrambling to Fill Empty Pipelines |
| | 4. China's Oil Importers Face Surprising Drought as U.S. Supplies Dry Up |
| | 5. China's Oil Consumption Plummets as U.S. Producers Fail to Break into the Market |

Table 8: Rephrasing, slight ablation, and negative modification of the headline "U.S. Oil Producers Find a Surprise New Market: China." Each augmentation displays the system prompt, user-provided headline, and model-generated responses listed with numbers.

### A.3 Quality Monitoring System for Augmentations

To ensure that the generated semantic augmentations align with the desired levels of semantic similarity, we employ a robust quality monitoring system. This system leverages a fine-tuned BERT model as a discriminator to validate the semantic relationships between base headlines and their augmentations. The primary objective is to confirm that the augmentations adhere to the predefined similarity thresholds associated with each augmentation action.

**1. Similarity Score Validation**   The discriminator model evaluates the similarity between a base headline and its augmented counterpart, producing a score in the range $[0, 1]$. These scores are compared against action-specific thresholds to classify the augmentations:

- **Reworded (Re):** Similarity scores must fall within the range $[0.66, 1.00]$, indicating high semantic alignment with minimal alteration in meaning.
- **Semantically-Shifted (S):** Scores between $[0.33, 0.66]$ reflect moderate semantic divergence while maintaining topical relevance.
- **Negated (N):** Scores in $[0, 0.33]$ denote significant semantic contrast or opposing meanings.

**2. System Workflow**   The process begins by passing the base and augmented headlines through the fine-tuned BERT model, which computes similarity scores using cosine similarity of their embeddings. These scores are then compared against the specified thresholds. If an augmentation fails to meet the desired threshold for its action type, it is flagged for review or discarded.

**3. Feedback Mechanism**   To iteratively refine the augmentation process, the quality monitoring system provides feedback to the generation pipeline. For instance, if a significant portion of reworded augmentations falls below the required threshold, the prompts for the augmentation model are adjusted, or additional constraints are imposed during headline generation.

**4. Ensuring Semantic Coherence**   The quality monitoring system serves a dual purpose: it enforces the semantic coherence of augmentations and ensures that the resulting augmented dataset aligns with the intended distribution of similarity scores. This guarantees that the augmented daily news sets (DNS) maintain the desired variability and semantic relationships required for effective contrastive learning.

This monitoring system plays a critical role in maintaining the integrity of the augmentation pipeline, thereby enhancing the reliability and utility of the ContraSim embedding space.

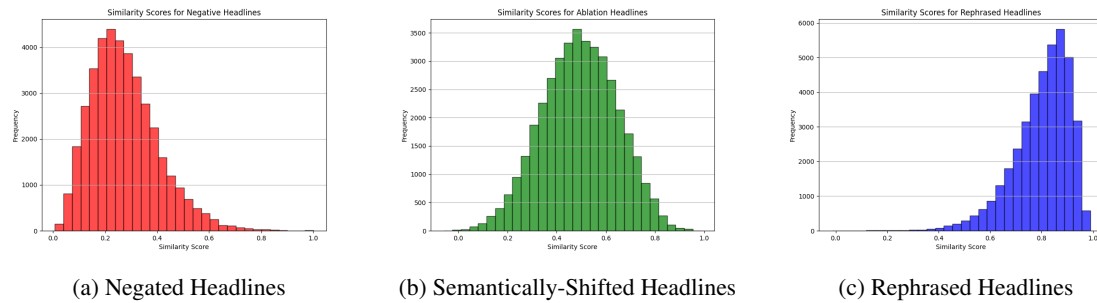

(a) Negated Headlines      (b) Semantically-Shifted Headlines      (c) Rephrased Headlines

Figure 2: Distribution of similarity scores for augmented headlines across different augmentation actions. Each histogram represents the frequency distribution of similarity scores produced by the quality monitoring system for a specific augmentation type: (a) Negated Headlines, showing a concentration of scores in the low similarity range ($[0, 0.33]$); (b) Semantically-Shifted Headlines, with scores distributed in the mid-range ($[0.33, 0.66]$); and (c) Rephrased Headlines, exhibiting high similarity scores ($[0.66, 1.00]$). These distributions validate that the augmentations align with their intended semantic similarity thresholds.

## B  SIMILAR HISTORY NEWS DAYS EXAMPLES

Below we outline a random daily news set from our test dataset, alongside the closest, the median, and the farthest daily news sets. We have colored headlines that we found semantically similar.

**Base Newsline:**
Carlos Ghosn to Stay in Jail After Tokyo Court Denies Bail Again
Apple Supplier Issues Profit Warning, Blames Weak Chinese Demand
Stocks Jump on Trade Hopes
New on Parents' To-Do List: Checking Kids' Credit History
Be Careful What You Wish For: Wall Street Hit by Fourth-Quarter Volatility
JPMorgan CEO Dimon's Compensation Tops Pre-Crisis Record
Sears Creditors Oppose Sale to Edward Lampert
PPG Says U.S. Attorney's Office Investigating Paint Giant's Accounting Practices
Netflix Surprises With Big Subscriber Gains, Shares Soar
In Canada, Oil Output Cuts Pay Off as Prices Rally
Bid to Create European Train Champion on Verge of Collapse
The U.S. and China Are Hatching a Plan to Reopen Chicken Trade
Midcap Stocks Try to Shed the 'Orphaned Index' Label
American Express More Than Doubles Annual Profit
U.S. Dollar Loses Earlier Gains
Oil Declines on Worries Over Rising U.S. Inventories
Bank of America's Wealth-Management Business Bucks Trend
BlackRock's Assets Fall Sharply
Gymboree Files for Chapter 11 Bankruptcy Protection Again
Germany Weighs Banning Huawei From 5G
John C. Bogle, Founder of Vanguard Group, Dies at 89
Moonves to Fight CBS Decision to Deny $120 Million Severance
As Americans Drink Less Alcohol, Booze Makers Look Beyond the Barrel
The Labor Market Is Strong, Despite Concerns About a Hiccuping Economy
Fed's Quarles Upbeat on U.S. Economy
Byron Trott's Firm Orders a Double, Adds Whiskey to Tequila Bet
Morgan Stanley Earnings Fall Short as Trading Hampers Results
Berkshire, NetJet Pilots Reach Labor Deal
OPEC Production Falls as Saudis Slash Output
PG&E Shareholder BlueMountain Protests Bankruptcy Decision
As China Slows, Sneakers and Noodles Beat Jewelry and Liquor Stocks

**Similarity Scores and Headlines:**

| Rank 1: | **Similarity: High** |
|---|---|
| | Headline: Judge Orders Sale of Citgo's Parent Unless Venezuela Posts Bond |
| | Gap Reports Sales Decline at Flagship Brand |
| | Stocks Jump on Trade Hopes |
| | Victoria's Secret Misses Out on Retailing's Surge |
| | U.S. Stocks Climb as Inflation Fears, Volatility Ebb |
| | Jamie Dimon Tops List of Highest-Paid Finance CEOs |
| | Gormley's Take: Why Some Startups Stay on the IPO Sidelines |
| | PPG Says U.S. Attorney's Office Investigating Paint Giant's Accounting Practices |
| | Netflix Reports Paid Customers Rise on Strength Overseas |
| | U.S. New Home Sales Fell in July |
| Rank 2: | **Similarity: Medium** |
| | BlackRock Plans to Block Walmart |

The Fine Print: What's in the Bank Deregulation Bill
GE to Merge Rail Division With Wabtec in $11 Billion Deal
Harker Says Fed Should Move Rates Up 'Judiciously'
Uber's New Rules on Arbitration for Sex-Related Claims Face First Test
Wells Fargo's True Culture Shines Through
SEC Program to Boost Small Stock Trading Hurt Market Quality
Oil Rises as Venezuela Election Puts Supply at Risk
State and Provincial Regulators Target Initial Coin Offerings
Camera Maker Arecont Vision Seeks Approval for Court-Supervised Sale

**Rank 3:**     **Similarity: Low**
New York Stock Exchange to Have First Female Leader in 226-Year History
Oil Keeps Rising Amid Worries About Venezuela, Iran Output
How to Launch Cryptocurrency Derivatives: CFTC Issues New Guidance
Diagnostics Firm Grail Raises $300 Million From China-Based Investors
Rising Dollar Sparks Tumult in Emerging Markets
Stocks Surge as Fears of a Trade War With China Ebb
Starbucks Clarifies Policy on Allowing Nonpaying Guests
Dollar Closes Lower Alongside Treasury Yields
Blackstone Unit Objects to Gibson Brands Financing
New Activist Is Dealt a Second Blow in HomeStreet Fight

## C    HOW AUGMENTATION ACTIONS AFFECT NEWS HEADLINE SIMILARITY

In this section, we investigate the effects of different augmentation strategies on news headline similarity within embedding space. The goal of this experiment was to quantify how rephrasing, semantic shifts, and negations impact the embedding distances of news headlines.

We began by selecting a dataset of daily news headlines, ensuring a diverse and representative sample of financial and general news topics. For each experiment:

1. **Two random days** were selected from the dataset.

2. **A headline** from one of these days was chosen as the base headline.

3. The chosen headline was subjected to one of the following augmentation actions using our algorithm:

    - **Rephrasing (Re):** Preserves the original semantic meaning but alters the phrasing.

    - **Semantic Shift (S):** Introduces slight changes in meaning while maintaining topic relevance.

    - **Negation (N):** Alters the meaning to convey the opposite sentiment or direction.

4. The base and augmented headlines were embedded into a semantic space using a pre-trained language model fine-tuned with Weighted Self-Supervised Contrastive Learning (WSSCL).

5. The change in embedding space distance was measured between the base and augmented headlines.

The average shifts in embedding distances, quantified as cosine similarity changes, were as follows:

- **Rephrased:** $+0.146$

- **Semantic-Shifted:** $+0.043$

- **Negated:** $-0.0642$

**Rephrased Headlines:** Rephrased headlines showed the largest positive shift in embedding distances ($+0.146$), indicating that while the phrasing varied, the core semantic content remained highly aligned. This demonstrates that rephrasing maintains the essence of the original headline, making it the most semantically consistent transformation.

**Semantic Shifts:** Semantic-shifted headlines exhibited a moderate positive shift ($+0.043$). This suggests that while some semantic information diverged, the augmented headlines still retained a level of topical similarity to the base headline. The variability in these distances reflects the subtle nuances introduced by the algorithm.

**Negated Headlines:** Negated headlines displayed a negative shift ($-0.0642$), indicating an intentional movement away from the base headline's meaning. This highlights the algorithm's capacity to generate semantically contrasting headlines effectively. The relatively small magnitude of this shift suggests that negation preserves certain structural or contextual elements, even when the semantic intent is inverted.

The results underline the versatility and precision of our augmentation strategies:

- **Rephrasing** can be leveraged for tasks requiring high semantic consistency.
- **Semantic Shifting** introduces controlled variability, useful for contrastive learning applications.
- **Negation** is effective for generating challenging counterexamples in adversarial tasks or for enhancing model robustness.

These findings validate the embedding model's sensitivity to nuanced semantic changes and demonstrate the utility of augmentation actions in crafting datasets for contrastive and supervised learning paradigms.

## D DATASETS

### D.1 NIFTY-SFT DATASET

The **N**ews-**I**nformed **F**inancial **T**rend **Y**ield (NIFTY) dataset Raeid et al. [2024] is a processed and curated daily news headlines dataset for the stock (US Equities) market price movement prediction task. NIFTY is comprised of two related datasets, NIFTY-LM and NIFTY-RL. In this section we outline the composition of the two datasets, and comment on additional details.

**Dataset statistics** Table 10 and Table 11 present pertinent statistics related to the dataset.

Table 10: Statistics and breakdown of splits sizes

| Category | Statistics |
| --- | --- |
| Number of data points | 2111 |
| Number of Rise/Fall/Neutral label | 558 / 433 / 1122 |
| Train/Test/Evaluation split | 1477 / 317 / 317 |

Table 11: Date Ranges of news headlines in splits

| Split | Num. Samples | Date range |
| --- | --- | --- |
| Train | 1477 | 2010-01-06 to 2017-06-27 |
| Valid | 317 | 2017-06-28 to 2019-02-12 |
| Test | 317 | 2019-02-13 to 2020-09-21 |

#### D.1.1 NIFTY-LM: SFT FINE-TUNING DATASET

The NIFTY-LM prompt dataset was created to finetune and evaluate LLMs on predicting future stock movement given previous market data and news headlines. The dataset was assembled by aggregating

Anticipate the direction of the $SPY by analyzing market data and news from 2020-02-06.

(a) Instruction component of a $\pi_{LM}$ policy query $x_q$.

date, open, high, • • •, pct_change, macd, boll_ub, boll_lb, rsi_30, • • •, close_60_sma

2020-01-27, 323.03, 325.12, • • •, -0.016, 2.89, 333.77, 319.15, 56.26, • • • , 317.40
2020-01-28, 325.06, 327.85, • • •, 0.0105, 2.59, 333.77, 319.55, 59.57, • • • , 317.78

• • •.      • • • •

2020-02-04, 328.07, 330.01, • • •, 0.0152, 1.3341, 333.60, 321.26, • • •, 319.41
2020-02-05, 332.27, 333.09, • • •, 0.0115, 1.7247, 334.15, 321.73, • • •, 319.82

(b) The market's **history** is provided as the past $t$ days of numerical statistics like the (OHLCV) price (in blue) and common technical indicators (in orange) (e.g. moving averages) data.

Figure 3: Breaking down the instruction or prompt prefix, and market context components of a prompt, $x_p$.

information from three distinct sources from January 6, 2010, to September 21, 2020. The compilation includes headlines from The **Wall Street Journal** and **Reuters News**, as well as market data of the $SPY index from **Yahoo Finance**. The NIFTY-LM dataset consists of:

- **Meta data**: Dates and data ID.

- **Prompt** ($x_p$): LLM question ($x_{question}$), market data from previous days ($x_{context}$), and news headlines ($x_{news}$).

- **Response**: Qualitative movement label ($x_r$) $\in \{Rise, Fall, Neutral\}$, and percentage change of the closing price of the $SPY index.

To generate LLM questions, ($\boldsymbol{x_{question}}$), the authors used the self-instruct Wang et al. [2023] framework and OpenAI GPT4 to create 20 synthetic variations of the instruction below:

> Create 20 variations of the instruction below.
> Examine the given market information and news headlines data on DATE to forecast whether the $SPY index will rise, fall, or remain unchanged. If you think the movement will be less than 0.5%, then return 'Neutral'. Respond with Rise, Fall, or Neutral and your reasoning in a new paragraph.

Where DATE would be substituted later, during the training phase with a corresponding date.

**Context** The key 'context' ($\boldsymbol{x_{context}}$) was constructed to have newline delimited market metrics over the past T ($\approx$ 10) days (N.B. Not all market data for the past days for were available and therefore prompts might have less than 10 days of market metrics.).

Table 12 show the details of financial context provided in each day's sample.

Table 12: Summary of the dataset columns with their respective descriptions.

| Column Name | Description |
| --- | --- |
| Date | Date of the trading session |
| Opening Price | Stock's opening market price |
| Daily High | Highest trading price of the day |
| Daily Low | Lowest trading price of the day |
| Closing Price | Stock's closing market price |
| Adjusted Closing Price | Closing price adjusted for splits and dividends |
| Volume | Total shares traded during the day |
| Percentage Change | Day-over-day percentage change in closing price |
| MACD | Momentum indicator showing the relationship between two moving averages |
| Bollinger Upper Band | Upper boundary of the Bollinger Bands, set at two standard deviations above the average |
| Bollinger Lower Band | Lower boundary, set at two standard deviations below the average |
| 30-Day RSI | Momentum oscillator measuring speed and change of price movements |
| 30-Day CCI | Indicator identifying cyclical trends over 30 days |
| 30-Day DX | Indicates the strength of price trends over 30 days |
| 30-Day SMA | Average closing price over the past 30 days |
| 60-Day SMA | Average closing price over the past 60 days |

**News Headlines** $(x_{news})$: Final list of filtered headlines from the aggregation pipeline. The non-finance related headlines were filtered out by performing a similarity search with SBERT model, "all-MiniLM-L6-v2" Reimers & Gurevych [2019]. Each headline was compared to a set of artificially generated financial headlines generated by GPT-4, with the prompt *"Generate 20 financial news headlines"*. Headlines with a similarity score below 0.2, were excluded from the dataset. To respect the prompting 'context length' of LLMs, in instances where the prompt exceeded a length of 3000 words, a further refinement process was employed. This process involved the elimination of words with a tf-idf Sammut & Webb [2010] score below 0.2 and truncating the prompt to a maximum of 3000 words.

It is also important to note that the dataset does not encompass all calendar dates within the specified time range. This limitation emanates from the trading calendar days, and absence of relevant financial news headlines for certain dates.

**Label** $(x_r)$: The label is determined by the percentage change in closing prices from one day to the next, as defined in equation 5. This percentage change is categorized into three labels: {Rise, Fall, Neutral}, based on the thresholds specified in equation 6.

$$PCT_{\text{change}} = \left( \frac{\text{Closing Price}_t - \text{Closing Price}_{t-1}}{\text{Closing Price}_{t-1}} \right) \times 100\% \tag{5}$$

$$x_r = \begin{cases} \text{Fall} & \text{if } PCT_{\text{change}} < -0.5\% \\ \text{Neutral} & \text{if } -0.5\% \leq PCT_{\text{change}} \leq 0.5\% \\ \text{Rise} & \text{if } PCT_{\text{change}} > 0.5\% \end{cases} \tag{6}$$

## D.2 BIGDATA22 DATASET

The BigData22 dataset is a comprehensive collection of financial tweets compiled between July 2019 and June 2020, designed to analyze the correlation between social media sentiment and financial market movements. It includes over 272,000 tweets distributed across 7,164 distinct trading days. Each tweet is annotated with one of two market movement labels: *Fall* or *Rise*, based on the performance of financial indices on the corresponding day.

BigData22 provides a unique perspective on market sentiment by focusing exclusively on social media platforms, contrasting with datasets like NIFTY-SFT that rely on curated headlines from reputable financial news outlets. While the reliance on social media introduces a higher degree of noise, it also brings diversity and real-time sentiment shifts into the dataset. This makes it an excellent benchmark for evaluating the robustness and adaptability of models like ContraSim, particularly in handling noisy, unstructured data with high variability.

Additionally, the dataset includes metadata such as timestamps and tweet authors, allowing researchers to explore temporal trends and user-specific sentiment biases. This temporal richness is especially valuable for studying dynamic sentiment patterns in financial contexts and assessing model performance in capturing short-term fluctuations influenced by social media activity.

## D.3 IMDB REVIEWS DATASET

The IMDB Reviews dataset is a widely used benchmark for sentiment analysis tasks, comprising 50,000 movie reviews, each accompanied by a sentiment score ranging from 0 to 10. The reviews are equally divided into training and testing sets, ensuring a balanced evaluation of model performance. For this project, the sentiment scores are grouped into three categories: *Low* (0.0-5.5), *Medium* (5.6-7.5), and *High* (7.6-10.0), creating a classification task to predict the overall sentiment of a review.

This dataset is particularly valuable for testing the generalizability of ContraSim beyond financial forecasting. Unlike financial datasets, which often feature structured news or market data, the IMDB dataset focuses on user-generated content with a wide range of writing styles and subjective expressions. By evaluating ContraSim on this dataset, we can assess its ability to adapt to orthogonal tasks, such as opinion mining and sentiment classification, that require capturing nuanced semantic relationships in text.

The diversity in review content, ranging from casual remarks to in-depth critiques, challenges the model to effectively distinguish between sentiment categories. This provides insights into ContraSim's capability to learn and represent global semantic structures across domains, making it a valuable tool for applications extending beyond finance, such as media analytics and customer feedback analysis.

