# OpenReview forum: "ContraSim: Contrastive Similarity Space Learning for Financial Market Predictions"
_ICLR.cc/2025/Conference — Submitted to ICLR 2025_

### Official Review · Reviewer_yAX5 · 2024-10-28

**Soundness:** 3
**Presentation:** 3
**Contribution:** 3
**Rating:** 6
**Confidence:** 3

**Summary:**

The paper introduces ContraSim, an innovative approach for predicting financial market movements that employs a two-part method: it first creates semantically varied financial headlines through Weighted Headline Augmentation, and then applies Weighted-Self Supervised Contrastive Learning (WSSCL) to refine the embedding space. This combination allows ContraSim to effectively group newslines that reflect similar market trends, achieving a notable 7% improvement in classification accuracy for financial forecasts. Additionally, the authors present a novel metric called Info-kNN to assess how well the embedding space captures significant semantic relationships within financial news.

**Strengths:**

It presents a methodology that integrates advanced techniques such as Weighted Headline Augmentation and Weighted-Self Supervised Contrastive Learning (WSSCL), providing a fresh perspective on the interplay between financial news and market behavior.  The introduction of the Info-kNN metric to evaluate semantic clustering in the embedding space is another key contribution, offering researchers a robust tool for assessing how well models capture the complexities of financial data.

**Weaknesses:**

First, the paper does not inform the reader how ContraSim contributes to the field relative to other approaches. Second, apparently LLMs bring their own biases to the model. Any biases inherited from the LLM can affect the quality of the data and predictions. Predictions based on headlines provides limited scope and prediction errors as headlines are usually written to take attention.

**Questions:**

What are the types of augmentations utilized?
What are the characteristics of the learned embedding space that influences the interpretability of financial predictions?
Can you elaborate the hyperparameter tuning process in more detail?

---

> ### Author Response · Authors · 2024-11-26
>
> **Dear Reviewer,**
>
> Thank you for your detailed review and for raising important points that help us improve our submission. Below, we address your concerns and questions in detail.
>
> ---
>
> ### Responding to Weaknesses
>
> > **[W1]** *The paper does not inform the reader how ContraSim contributes to the field relative to other approaches.*
>
> This is an excellent point. In the revised version of the paper, we have added a discussion of how ContraSim contributes to the field. Specifically, we demonstrate that the self-supervised approach adds learns about market condition structure without requiring explicit market data. This makes it especially valuable for cases with fuzzy or incomplete ground truth values. Furthermore, we show that ContraSim consistently enhances the performance of downstream classification models. This improvement underscores its potential for financial forecasting models that rely on news headline data or any set of daily textual information.
>
> ---
>
> > **[W2]** *LLMs bring their own biases to the model. Any biases inherited from the LLM can affect the quality of the data and predictions. Predictions based on headlines provide limited scope and prediction errors as headlines are usually written to attract attention.*
>
> We agree that the use of LLMs introduces potential biases, and this is a limitation we acknowledge in our work. However, in financial market prediction, incorporating textual data is increasingly important as markets are influenced by human factors that cannot be fully captured by numerical indicators alone. We utilize the WSJ headlines dataset as an approximation of a knowledge base on important global events that may affect market conditions.
>
> To address this concern, we conducted an additional ablation study using Twitter data, which offers a different perspective on daily events. While Twitter data may also contain biases and sensationalism, it provides a broader and more diverse set of opinions. This new analysis has been included in the revised submission, and we believe it highlights the robustness of our method across different text corpora.
>
> ---
>
> ### Responding to Questions
>
> > **What are the types of augmentations utilized?**
>
> We use semantic augmentations generated via a fine-tuned LLAMA model. These augmentations include:
> 1. **Rewording:** Generating semantically equivalent headlines with different phrasing.
> 2. **Semantic Shifting:** Modifying the headline to slightly alter its semantic meaning while retaining relevance.
> 3. **Negation:** Transforming a positive headline into its negative counterpart and vice versa.
>
> These augmentations are only used during the training phase to enrich the self-supervised learning process and are discarded during inference. Details on the augmentation process are included in the appendix of the revised submission.
>
> > **What are the characteristics of the learned embedding space that influence the interpretability of financial predictions?**
>
> The embedding space created by our WSSCL approach captures semantic relationships between headlines. This allows for clustering similar market conditions and identifying patterns across historical data. Analysts can use this similarity space to contextualize current market trends with analogous past events, offering an interpretable framework for decision-making.
>
> > **Can you elaborate on the hyperparameter tuning process in more detail?**
>
> We agree that hyperparameter tuning is a critical aspect of our framework. In Section 4.4 of the revised paper, we provide a comprehensive summary of our tuning process.
>
> ---
>
> We thank you again for your thoughtful feedback, which has been invaluable in improving our work. We hope that the revisions address your concerns and provide the clarity and rigor you were seeking.

---

### Official Review · Reviewer_LYzi · 2024-10-30

**Soundness:** 3
**Presentation:** 3
**Contribution:** 3
**Rating:** 6
**Confidence:** 4

**Summary:**

The paper presents ContraSim, a contrastive learning framework for financial market prediction that creates a dense embedding space for financial news headlines. The approach has two main components:

1. Weighted Headline Augmentation: This technique generates variations of financial headlines with known semantic distances, including reworded, slightly ablated, and negative versions, allowing the model to capture rich semantic relationships between headlines.
2. Weighted Self-Supervised Contrastive Learning (WSSCL): Extending traditional contrastive learning, WSSCL uses augmented headlines to establish a continuous similarity space, clustering semantically similar headlines and separating dissimilar ones.

The framework introduces a novel metric, Info-kNN, to measure the density of semantically similar clusters. When integrated with a language model-based classifier, ContraSim achieves a 7% improvement in classification accuracy and a 13% boost in balanced accuracy over the baseline. Additionally, it provides practical support for financial analysts by identifying similar historical market days, offering valuable insights for forecasting.

**Strengths:**

1. Innovative and Practical Framework: The integration of contrastive learning through Weighted Headline Augmentation and WSSCL in financial market prediction presents a novel approach that assists financial analysts by enabling the identification of historical market conditions.
2. Enhanced Performance: The framework’s substantial improvements in classification accuracy (+7%) and balanced accuracy (+13%) validate its effectiveness.
3. Info-kNN Metric: The introduction of the Info-kNN metric, which evaluates clustering in the embedding space using information-theoretic principles, is a notable contribution. This metric provides a new way to assess contrastive learning models and has potential applications beyond financial data.

**Weaknesses:**

1. Limited Ablation Studies: The current ablation studies are limited, particularly in assessing the individual impact of the headline augmentations (reworded, slightly ablated, and negative). Examining how each augmentation affects accuracy and clustering—especially by showing the effect of their removal—would strengthen the claims. A similar analysis could be extended to Info-kNN and WSSCL components.
2. Narrow Scope in Experiments: Although the authors mention plans for broader application, the current validation is confined to financial data, which limits the immediate generalizability of the findings. Including experiments from other domains or providing a more in-depth discussion of potential applications would enhance the broader impact of this work.

**Questions:**

Could you provide more detailed insights into how the different headline augmentation types (reworded, slightly
ablated, negative) contribute individually to the observed improvements in classification accuracy and clustering
effectiveness? While the paper outlines these augmentations, it would be helpful to understand the individual
impact of each augmentation type on the model's learning process. Additionally, a more detailed ablation study
of key components, such as augmentation types, Info-kNN, and WSSCL, would offer a clearer understanding of
their respective contributions to the overall performance.

---

> ### Author Response · Authors · 2024-11-26
> **Re: On Increased Ablations**
>
> Dear Reviewer,
>
> Thank you for the detailed review, and for your acknowledgment of the strengths of our work.
>
> ---
>
> ### Responding to Questions and Weaknesses
>
> > **[W1]** *Limited Ablation Studies: The current ablation studies are limited, particularly in assessing the individual impact of the headline augmentations (reworded, slightly ablated, and negative). Examining how each augmentation affects accuracy and clustering—especially by showing the effect of their removal—would strengthen the claims. A similar analysis could be extended to Info-kNN and WSSCL components.*
>
> We appreciate your emphasis on the importance of understanding the individual contributions of each augmentation type. In response, we are expanding the ablation studies to include an analysis of how removing each augmentation—reworded, slightly ablated, and negative—affects accuracy and clustering metrics. These results will be included in the appendix, alongside a detailed evaluation of Info-kNN and WSSCL components. We will notify you once this section is finalized and included in the revised version.
>
> ---
>
> > **[W2]** *Narrow Scope in Experiments: Although the authors mention plans for broader application, the current validation is confined to financial data, which limits the immediate generalizability of the findings. Including experiments from other domains or providing a more in-depth discussion of potential applications would enhance the broader impact of this work.*
>
> We agree that broadening the scope of experiments is crucial for demonstrating the generalizability of our approach. To address this, we have incorporated two additional datasets:
>
> 1. **BigData22 Financial Dataset:** We applied ContraSim to a corpus of tweets instead of WSJ headlines, providing insights into its performance on a different type of financial data.
>
> 2. **IMDB Dataset:** To explore a non-financial domain, we used ContraSim on movie reviews to predict movie ratings. This experiment demonstrates the framework's applicability to textual data outside finance and highlights its potential for broader adoption.
>
> These results and discussions are now included in the updated submission. We look forward to your feedback on this extended analysis.
>
> ---
>
> We thank you again for your thoughtful comments, which have helped us strengthen the paper. We are committed to addressing these points thoroughly and look forward to your insights on the revised version.

---

> > ### Author Response · Authors · 2024-11-28
> > **Re: On Increased Ablations**
> >
> > Dear Reviewer,
> >
> > We sincerely appreciate your insightful feedback, which has greatly helped improve our work. We are looking forward to **your thoughts on the updated manuscript**, particularly the expanded results in Section `S4.2: Results [Ln. 397]`, where we have incorporated findings from the original dataset and two additional datasets, as per your valuable suggestions.
> >
> > Your input will be instrumental in refining the manuscript further, and we look forward to hearing your thoughts.
> >
> > Respectfully,
> >
> > The Authors

---

### Official Review · Reviewer_kiCt · 2024-11-03

**Soundness:** 1
**Presentation:** 1
**Contribution:** 1
**Rating:** 3
**Confidence:** 5

**Summary:**

This paper proposes a model for stock movement prediction.

**Strengths:**

N/A

**Weaknesses:**

1. Low-quality presentation.
2. Lack of serious experiments.
3. Lack of contributions.

**Questions:**

The presentation of this paper is poor, which directly affects readers' understanding of the model design and problem formulation. It is unclear whether some model details are omitted or buried under misleading representations
1. There are numerous grammar mistakes and typos in the first paragraph of Section 3.1, with almost every sentence containing issues. This makes understanding the motivation behind building the proposed embedding space difficult.
2. It is unclear why the authors chose to use 10 and 30 to control the number of news articles sampled per day. Are these numbers theoretically justified, suggested by domain experts, or supported by empirical results? Also, what happens if there are fewer than 10 financial-related news articles in a day?
3. What is the motivation behind using random sampling when there are more than 30 news articles per day? The use of randomness makes it difficult to ensure that the space will cover all topics described in the news for that day. In the worst-case scenario, important topics might be completely omitted.
4. The authors claim that they only keep newslines and ignore tabular data. What does this mean? This statement seems to come out of nowhere.
5. In the description of creating headlines, it is unclear why "prompts" are sometimes created and, at other times, "headlines" are created. Is there a specific reason for this, or are these typos?
6. In Equation 3, it is unclear what the text refers to.
7. The notation $N_i$ in the description below Equation 3 should be $N_k$
8. The symbol $h$ in Equation 3 should be $\hat{h}$
9. The augmentation method lacks a quality control mechanism for headline generation. Also, the definitions of "slight ablation" and "negative" are unclear. In the given example, it appears that the authors only used a language model to generate fake company names, which lacks a clear motivation.
10. The section on "generating newsline buckets" is difficult to understand due to grammar mistakes and undefined terms, making the entire subsection challenging to follow.
11. The method used by the authors to achieve "known semantic distance" lacks motivation. It appears that the authors manually assigned values of 1, 0.5, and 0 to three different augmentations without justification.
12. Grammar mistakes and typos continue throughout the rest of the paper, making it challenging to list them all. However, there are some critical ones. For example, in the WSSCL section, the authors state that there are only three movements: Fall, Natural, and Fall again.
13. The experiment section lacks serious comparisons to justify the performance of the proposed model.

**Details Of Ethics Concerns:**

The authors used an LLM to augment news headlines without any constraints. In the example provided, some of the generated headlines are misleading and fake.

---

> ### Author Response · Authors · 2024-11-26
> **Re: Author Responses to kiCt - Summary and Raised Ethical Review Flag [1/n]**
>
> Dear Reviewer,
>
> We sincerely appreciate the time and effort you have taken to provide detailed feedback on our manuscript. Your comments have been instrumental in helping us refine and improve the work. Below, we address your concerns in detail, categorized by the specific issues raised.
>
> ### Response to Summary:
> > This paper proposes a model for stock movement prediction.
>
> We respectfully clarify that this is a mischaracterization of our work. Our work proposes a **contrastive self-supervised learning technique** whose efficacy was demonstrated (initially) using US financial market movement prediction, and (subsequently, i.e. now) using varied asset classes movement and IMDB movie sentiment analysis. It is by no means, **a model for stock movement prediction**.
>
> ### Response to the Raised Ethical Review Flag
>
> Although we appreciate your diligence in identifying potential ethical concerns with the use of LLMs, we respectfully find the ethical review flag for our work misplaced. While it is true that we employed an LLM to create augmented news headlines, these augmented headlines were used exclusively within a _self-supervised learning_ paradigm. We use LLMs to reword, slightly modify the meaning of, and negate headlines in the corpus of WSJ headlines as a method of _self-supervised contrastive learning_. Importantly, **all augmented headlines are discarded immediately after training**.
>
> Once deployed, there are absolutely no LLM generated headlines used at all. We utilize our contrastive embedding space to measure the similarity between real headlines from the Wall Street Journal. This methodology mirrors standard practices in self-supervised learning (e.g., image clustering or data augmentation) and is entirely consistent with common techniques adopted in the post-LLM era.
>
> Drawing from Paul Bosanac's magnum opus [1], we argue that ethical reasoning must be grounded in consistent principles. Applying the ethical flag here would necessitate flagging all analogous work using self-supervised learning, creating an unnecessary and restrictive standard. Specifically:
>
> > If this work is flagged as ethically concerning solely for using LLM-generated augmentations in training, then by extension, all research leveraging self-supervised learning with data augmentation would warrant similar flags. Such a precedent would severely restrict innovation in domains that widely rely on these techniques.
>
>
>
>
> [1] As explained in Paul Bosanac, "Litigation Logic: A Practical Guide to Effective Argument." (2009)

---

> ### Author Response · Authors · 2024-11-26
> **Re: Author Responses to kiCt [2/n]**
>
> ### Response to weaknesses (and associated) questions
>
> > **[W1 + Qs: 1, 5, 6, 7, 8, 10, 12] Low-quality presentation**:
>
> Firstly, we thank you for noting that the presentation quality was not up to par. While we recognize that this is a subjective notion, we took your characterization seriously. We frankly agree that the certain presentation aspects (like grammar mistakes, results presentation etc.) had room for improvement. Thus, we worked really hard to address these issues earnestly in our updated manuscript version. We hope you’ll find that the stylistic and presentation lackings should be now addressed. We look forward to your thoughts after your kind review of the updated script.
>
> Key improvements include:
> - Revising Sections 2, 3 to clarify the motivation and eliminate ambiguities, and streamline the methods.
> - Streamlining mathematical notations, especially in equations for brevity and consistency (Section 3.1 and 3.2).
> - Updated Table 1 to elucidate transformation actions better with an example.
> - Updated Figure 1 to better illustrate the full ContraSim pipeline.
> - Added Appendix sections A. Headline Transformations, B. News Headline Similarity Examples, C. Measuring Information Density, D. Datasets, for added clarity on details of entire ContraSim pipeline.
>
>
> Thank you for pointing out typographical mistakes, typos etc. under questions.
> Please note that questions 1, 5, 6, 7, 8, 10, and 12 are typographical mistakes and have been addressed in our revised manuscript.
>
> ---
>
> > **[W2] Lack of Serious Experiments**:
>
> We appreciate your feedback and have expanded our experimental evaluation significantly in the revised version. We request you to kindly review the global response [above](https://openreview.net/forum?id=GfuJR76Sfo&noteId=LsuxRan2Mg).
>
>
> To summarize:
> - Experiments now include additional datasets from different domains (e.g., financial news from WSJ, financial news from social media, and movie reviews on IMDB) to demonstrate the versatility of our method.
> - Efficacy of ContraSim evaluated across a classification task (Table 4) and an information density approach (Table 5).
> - A detailed description of the design methodology and how the model was optimized are provided in Appendices A, B, and D.
>
> ---
>
> > **[W3] Lack of Contributions**:
>
> Our contributions are as follows:
> - A novel contrastive self-supervised learning framework tailored for extracting meaningful embeddings in problems where a collection of noisy features influence a single global feature. Here we explore how we can use financial news headlines to predict a single global market condition (Table 4 NIFTY-SFT, BigData22). We show the generalizability of this approach through using independent movie reviews to predict a global movie rating (Table 4 IMDB).
> - A practical augmentation strategy for news headlines with a known semantic similarity to the original (See Section 3.2). Using this augmentation strategy a self-supervised learning task is presented that groups headlines with similar semantic meaning closer together. In deployment, we can then find which real headlines are most similar to each other.
> - By leveraging an information density approach, we demonstrate that the ContraSim pipeline inherently clusters news headlines corresponding to days with homogenous market movement (Table 5). This indicates that ContraSim can effectively learn the dynamics influencing market direction without explicit supervision.

---

> ### Author Response · Authors · 2024-11-26
> **Re: Author Responses to kiCt [3/n]**
>
> ### Responses to Questions
>
> > [Q1, 5, 6, 7, 8, 10, 12] Presentation-Related Issues
>
> As outlined earlier, we appreciate you pointing these out, and they have been addressed in our updated manuscript.
>
> > [Q2, 3] On the Number of Headlines in in the News Headline Batch
>
> You correctly pointed out the semi-arbitrary number of headlines per newsline batch, we used in our self-supervised method. Previously, we required the number of newslines to be between 10 and 30. We have run experiments removing this bound, and generalizability was not lost.
>
> > [Q4] “The authors claim that they only keep newslines and ignore tabular data. What does this mean? This statement seems to come out of nowhere.”
>
> In this statement we were differentiating between composite models that use financial indicators (daily stock movement, volume, etc.) and headline data, and our approach which is interested only in headlines. This has been made more clear.
>
> > [Q9] “The augmentation method lacks a quality control mechanism for headline generation. Also, the definitions of "slight ablation" and "negative" are unclear. In the given example, it appears that the authors only used a language model to generate fake company names, which lacks a clear motivation.”
>
> This question was very insightful. You were correct in questioning the quality control mechanisms in the generation of headlines. To solve this we added a semantic control mechanism using a pretrained BERT semantic similarity model that acts as a quality control tool. We require rephrased augmentations to have a high similarity score, slightly ablated (now referred to as semantically-shifted) prompts require middle similarity scores, and negative must have low. This control mechanism also has a dual purpose in that it better defines the definitions of what a rephrased, semantically-shifted, and negative is.Details of the implementation are to be outlined in Appendix D.
>
> Additionally, fake company names as an artifact of the augmentation pipeline are an unfortunate byproduct of the self-supervised approach we propose. However, we find that there is minimal impact on downstream performance because the nature of the self-supervised task is to cluster on the basis of clustering headlines on headline semantic similarity.
>
> > [Q11] "The method used by the authors to achieve "known semantic distance" lacks motivation. It appears that the authors manually assigned values of 1, 0.5, and 0 to three different augmentations without justification."
>
> In the revised manuscript, we have removed the use of term semantic distance. Semantic distance was defined as 1 - semantic similarity. Furthermore, we have strengthened the definition and motivation of the semantic similarity metric section 3.2, and specifically how semantic similarity is used in the loss functions proposed.
>
> > [Q13] The experiment section lacks serious comparisons to justify the performance of the proposed model.
>
> Refer back to [W3]

---

### Official Review · Reviewer_LVGH · 2024-11-14

**Soundness:** 2
**Presentation:** 3
**Contribution:** 2
**Rating:** 5
**Confidence:** 4

**Summary:**

In this paper, a novel Weighted Self-Supervised Contrastive Learning (WSSCL) method is introduced to cluster news-lines based on learned semantic embeddings. LLMs are used to create modified prompts containing semantically identical or augmented news-lines. An enhanced version of KNN algorithm based on information theory is also introduced for clustering, in order to handle imbalanced labelled classes. Experiments on the task of stock market direction prediction show the effectiveness of the proposed method.

**Strengths:**

The paper focuses on an interesting problem, that is stock market direction prediction. The proposed method based on contrastive learning of text embeddings is reasonable. The example about rephrasing, slight ablation, and negative modification of the headline in Table 1 is illustrative and to the point.

**Weaknesses:**

Only one dataset (the NIFTY dataset) is used. It'd be great if the authors could use more datasets for evaluation.

**Questions:**

The estimated baseline values are a mean of random samples following the (23%, 60%, 17%) label split in Table 3. Does it mean if a trivial classifier always predicts the label of a sample is Neutral, it would achieve 60% accuracy?

---

> ### Author Response · Authors · 2024-11-26
> **Re: Addressing the Dataset Limitation**
>
> Dear Reviewer,
>
> Thank you for acknowledging the strengths of our paper and for your valuable feedback and review.
>
> ### Response to Weakness
>
> > **[W1]** Only one dataset (the NIFTY dataset) is used. It'd be great if the authors could use more datasets for evaluation.
>
> Thank you for pointing out this weakness. We agree that adding more datasets would indeed make our work more empirically robust. As such, we have significantly improved our results section by adding 2 additional datasets, one from financial domain but with different headlines content and asset classes, and the second from an orthogonal domain and task: the well-known IMDB movie review dataset.  We have added further details in our global response above.
>
> ### Response to Questions
>
> > **[Q1]** The estimated baseline values are a mean of random samples following the (23%, 60%, 17%) label split in Table 3. Does it mean if a trivial classifier always predicts the label of a sample is Neutral, it would achieve 60% accuracy?
>
> It’s important to make distinction between clustered accuracy vs. global accuracy. You don’t have access to global accuracy.
> Question answer: In Table 4), the accuracy follows a clustered approach. It is K-Means accuracy within the closest 5 nodes. Therefore, a naive model cannot necessarily predict 60% accuracy global when it can only access local information. For Table 3), we balance the datasets to be 33, 33, 33. So baseline accuracy from a naive model can only be 33%.
>
>
> Please find these added datasets and results in our updated `Results` section in added  tables   4, 5.
>
> We look forward to your feedback regarding our these added experiments and results.

---

> > ### Author Response · Authors · 2024-11-28
> > **Re: Re: Addressing the Dataset Limitation**
> >
> > Dear Reviewer,
> >
> > We are eagerly **looking forward to your feedback** on our updated manuscript. Specifically, section `S4.2 Datasets` **[Ln. 372]** contains descriptions of the three datasets used, updated with two additional datasets based on your helpful feedback.
> >
> > Respectfully,
> >
> > The Authors

---

### Author Response · Authors · 2024-11-25
**[Update] Added Experiments to Address Dataset Limitation and Empirical Scope**

We thank the reviewers for the thoughful reviews and suggestions to improve our manuscript. One common weakness suggested by multiple reviewers  (`LVGH` and `LYzi`) was the lack of additional datasets and/or any evaluation for generalizability (broader application) of our proposed approach and/or need of more experiments (`kiCt`) across other tasks/domains.

### Updated (Additional) Datasets
To address this limitation, we have now conducted experiments on two additional datasets: i) BigData22 (which is another financial news headlines dataset, but contains many different asset classes and tickers, unlike the NIFTY dataset that tracks the overall S&P500 market), and ii) the popular IMDB movie review classification dataset - as the name implies, is an orthogonal dataset/experiment in terms of domain and task.

| **Name of the Dataset** | **Problem Domain**     | **Headlines** | **Days/Movies** | **Date Range**              |
|--------------------------|------------------------|---------------|-----------------|-----------------------------|
| NIFTY-SFT               | Financial Headlines   | 106,271       | 696             | 2014-01-02 to 2015-12-30    |
| BigData22               | Financial Tweets      | 272,762       | 362             | 2019-07-05 to 2020-06-30    |
| IMDB Review             | Movie Reviews         | 955,788       | 352             | 2017-01-03 to 2017-12-28    |

### Updated Experiment Results

Our manuscript is now updated with the results of these additional experiments (in Section 4). These addition results further bolster the efficacy of our proposed approach.

---

| **Model**        | **NIFTY-SFT (Accuracy / F1)** | **BigData22 (Accuracy / F1)** | **IMDB (Accuracy / F1)** |
|-------------------|-------------------------------|--------------------------------|---------------------------|
| Baseline         | 0.3333 / 0.3333              | 0.5000 / 0.5000               | 0.3333 / 0.3333          |
| $Class_{Proj}$   | 0.3434 / 0.3389              | 0.5005 / 0.5016               | 0.3900 / 0.3897          |
| $Class_{LLM}$    | 0.3522 / 0.3833              | 0.5150 / **0.5094**           | 0.4518 / 0.4124          |
| $Class_{Both}$   | **0.3774** / **0.4670**      | **0.5156** / _0.5089_         | **0.5198** / **0.4620**  |

Table 4: Ablation study on market movement prediction, comparing classification models using ContraSim embedding vectors. $Class_{Proj}$ relies solely on ContraSim embeddings for predictions, $Class_{LLM}$ uses an LLM fine-tuned on market movement tasks, and $Class_{Both}$ combines both features. Results highlight that $Class_{Both}$ consistently outperforms other configurations across datasets. Notably, $Class_{Both}$ achieves the highest Accuracy and F1 scores on all datasets, including the IMDB task, where movie ratings (low, medium, high) are predicted using reviews. This demonstrates the effectiveness of combining similarity space projections with LLM embeddings for diverse classification tasks.

| **Dataset**     | **Model**             | **Info-KNN (k=5)** | **KNN (k=5)** | **KL-Divergence** | **JSD**  |
|------------------|-----------------------|--------------------|---------------|-------------------|----------|
| **NIFTY-SFT**   | Baseline              | .5916              | .4668         | .3539             | .1054    |
|                  | $\mathcal{L}_{CWCL}$ | .7647              | .4732         | **.3821**         | **.1164**|
|                  | $\mathcal{L}_{WSCL}$ | **.7219**          | **.5205**     | .3740             | .1144    |
| **BigData22**    | Baseline              | .7951              | .5506         | .1499             | .0452    |
|                  | $\mathcal{L}_{CWCL}$ | **.9084**          | **.7101**     | .2030             | .0607    |
|                  | $\mathcal{L}_{WSCL}$ | .8590              | .5507         | **.2246**         | **.0640**|
| **IMDB**         | Baseline              | .7456              | .5781         | .2919             | .0818    |
|                  | $\mathcal{L}_{CWCL}$ | .7626              | **.7500**     | **.3957**         | **.1120**|
|                  | $\mathcal{L}_{WSCL}$ | **.8252**          | .6875         | .3024             | .0908    |

Table 5: Results demonstrate that ContraSim embeddings effectively cluster real WSJ headlines (NIFTY-SFT) and Financial Twitter posts (BigData22) based on market movement directions (falling, neutral, rising) without explicit ground truth labels. Additionally, a similar clustering effect is observed with movie reviews, where reviews of movies with similar ratings (low, medium, high) are grouped together.

---

We would be most grateful if the reviewers could kindly review the update version w/ these significant additional experiments to (we hope, conclusively) address the main limitation/concern raised, and update the overall score of the paper if they are satisfactory.

Sincerely,

the Authors.

---

### Meta-Review · Area_Chair_FFup · 2024-12-16

**Metareview:**

This paper proposed a novel Weighted Self-Supervised Contrastive Learning (WSSCL) method for financial market prediction. The proposed framework introduces a novel metric, Info-kNN, to measure the density of semantically similar clusters. Evaluations on a dataset show the food performance of the proposal. The reviewers have concerns on the insufficient novelty of the paper, the weak evaluation, and the lack of clear explanations on ContraSim's contribution.

**Additional Comments On Reviewer Discussion:**

The  authors have provided rebuttals, but no reviewers would change their scores.

---

### Decision · Program_Chairs · 2025-01-22

Reject